# Inter-Rater Reliability of Ergonomic Work Demands for Childcare Workers Using the Observation Instrument TRACK

**DOI:** 10.3390/ijerph17051607

**Published:** 2020-03-02

**Authors:** Malene Jagd Svendsen, Peter Fjeldstad Hendriksen, Kathrine Greby Schmidt, Mette Jensen Stochkendahl, Charlotte Nørregaard Rasmussen, Andreas Holtermann

**Affiliations:** 1Musculoskeletal Disorders and Physical Work Demands, National Research Centre for the Working Environment, Lersø Parkallé 105, 2100 Copenhagen Ø, Denmark; peter.r.hendriksen@gmail.com (P.F.H.); kgs@nfa.dk (K.G.S.); cnr@nfa.dk (C.N.R.); aho@nfa.dk (A.H.); 2Institute of Sports Science and Clinical Biomechanics, University of Southern Denmark, Campusvej 55, 5230 Odense M, Denmark; m.jensen@nikkb.dk; 3Nordic Institute of Chiropractic and Clinical Biomechanics, Campusvej 55, 5230 Odense M, Denmark

**Keywords:** inter-rater reliability, ergonomics, work demands, observation

## Abstract

The aim of this study is to evaluate the inter-rater reliability of a newly developed instrument-TRACK (observaTion woRk demAnds Childcare worK) for observations of ergonomic work demands in childcare work. Two trained raters conducted thirty hours of concurrent observation of fifteen childcare workers in three different day nurseries. Inter-rater reliability of ergonomic work demands was evaluated using Gwet’s Agreement Coefficient (AC_1_) and interpreted by the Landis and Koch benchmark scale. Twenty ergonomic work demand items were evaluated. Inter-rater reliability was ‘almost perfect’ for nine items (AC_1_ 0.81–1.00), ‘substantial’ for four items (AC_1_ 0.61–0.80), ‘moderate’ for four items (AC_1_ 0.41–0.60), ‘fair’ for two items (AC_1_ 0.21–0.40), and ‘slight’ (AC_1_ 0.00–0.20) for one item. No items had ‘poor’ (AC_1_ < 0.00) agreement. The instrument is reliable for assessing ergonomic work demands in childcare in real-life settings.

## 1. Introduction

Childcare workers in Danish day nurseries take care of children age 0–3 years. Childcare workers report high physical workload and levels of physical exertion during work, and high prevalence of musculoskeletal pain and sickness absence rates [1]. These work-related exposures and consequences have been attributed to the childcare workers’ prioritisation of the safety and comfort of the children over the physical work environment [2,3]. As a result, the ergonomic exposures in childcare work include lifting and carrying of children, awkward postures or uncomfortable working positions, bending, pushing and pulling as well as sitting on the floor and on small furniture intended for children [2,3,4]. 

Observational instruments have been developed and applied to identify and quantify ergonomic work exposures or risk factors, with metrics including e.g., postures, body segments, risk levels, perceived exertion or discomfort, job profiles and load etc., through direct observational methods, self-reporting, or direct objective measurement methods—or a combination of these [5,6,7,8,9,10,11,12,13,14,15,16,17]. Direct objective measurements are often favoured over observation and self-report [18] as self-report is known to be imprecise and potentially biased [19,20,21,22]. However, direct objective measurements are equipment- and personnel-intensive and expensive [15,18], and direct observation methods with video recording furthermore presents an ethical dilemma for privacy reasons.

For childcare work, the existing instruments for identification and quantification of ergonomic work exposures is, to the best of our knowledge, limited to self-report [23,24] or direct objective measurements occasionally with simultaneous video recordings [25,26,27,28,29]. No direct observational instrument has been developed specifically for childcare work. Therefore, the observaTion woRk demAnds Childcare worK (TRACK) observational instrument was developed for the Danish TOY-project to assess ergonomic work demands in day nurseries for live observations using a handheld tablet. TRACK was developed based on a modified Task Recording and Analysis on Computer-approach [30,31]. The TOY-project aims at reducing physical exertion, musculoskeletal pain, and pain-related work interference through a participatory ergonomic workshop intervention with baseline and follow-up measurements of ergonomic work demands. The instrument was designed to primarily evaluate ergonomic work exposures that we currently are not able to measure with accelerometers, such as lifting, carrying and squatting and to give more detailed information than accelerometers allow for some exposures [1]. The instrument was also designed to determine during which childcare specific tasks or situations the ergonomic work demands occurred in.

Development of a new observation instrument should entail reliability testing of the method [6] as poor reliability of an instrument affects validity and eventually correct interpretation [32]. The purpose of this study was to evaluate the inter-rater reliability of the TRACK observational instrument during normal childcare work. Specifically, we assessed the agreement of observations of ergonomic work demands conducted by two raters in terms of the occurrence, synchronised timing and duration.

## 2. Materials and Methods 

### 2.1. TRACK Observation Tool

The TRACK instrument was developed to systematically record observations of ergonomic work demands. To identify typical childcare work situations and ergonomic work demands in day nurseries, four researchers visited six day nurseries to observe childcare workers’ workdays. The observations were combined with information from the Danish Work Environment Authority about ergonomic exposures in childcare to form the basis of TRACK. Two work environment consultants from the municipality of Copenhagen also gave input to the content. A beta version of the observation instrument with categories of different ergonomic work demands was created and pilot tested in two day nurseries by four researchers. This led to minor changes. A procedure manual for observations and definitions of each item in the instrument was also prepared.

#### Description of TRACK 

The TRACK instrument contained 20 items within four categories: setting, ergonomic exposures, situations and disturbances as presented in Table 1. Some items had a descriptive factor that modified items into sub-items, providing more detailed information. If an item had descriptive factors, at least one factor had to be registered. One setting and one situation always had to be selected, while ergonomic exposures and disturbances were registered only when occurring. The occurrence of an item, i.e., an event, was registered as either a ‘state event’ or ‘point event’. A ‘point event’ was registered at a single time point, providing information of the instantaneous occurrence only, while a ‘state event’ was registered over time, providing information of both onset and duration (referred to as entity types, see Section 2.3.1 and Figure 1). Duration of ‘state events’ was recorded in one of the two ways: (1) Recording both starting and stopping time manually (referred to as ‘non-exhaustive’) or (2) recording the starting time manually, but the stopping time automatically when a new item within the same category was registered (referred to as ‘mutually exclusive and exhaustive’). The instrument was intended for sampling continuously in time, making it possible to record both durations and frequencies for registered items. The definition of an observation in this study was the continuous sampling from time 0:00 to 120:00 min. The instrument was developed using the Observer XT software (The Observer XT version 14; Noldus Information Technology, Wageningen, The Netherlands) and subsequently transferred for use in the Pocket Observer software (Pocket Observer version 3.3.46; Noldus Information Technology, Wageningen, The Netherlands) on handheld tablets (GT-P3100 or SM-T280; Samsung, Suwon, South Korea) for convenient field observations. 

### 2.2. Reliability Evaluation

To evaluate the reliability, 30 h of field observations of childcare work was conducted by two raters simultaneously.

#### 2.2.1. Study Population and Sample Size

Day nurseries were eligible for participation in the reliability evaluation if they were situated in the municipality of Copenhagen and employed a minimum of nine childcare workers. Day nurseries that had participated in the randomised controlled trial (RCT) (the TOY-project) [1] were not eligible. All childcare workers employed in the day nurseries were eligible as ‘objects’ for inter-rater observations. The nurseries were incentivised for participation by being offered a condensed 1.5 h version of the intervention workshops from the TOY project. Prior to contacting eligible nurseries, the required number of observation hours for the inter-rater reliability evaluation was calculated using observational data from the RCT, which was conducted prior to the reliability evaluation [1]. The calculation of hours needed was contingent on a Poisson distribution of the observational data as the probability of an item was limited to occurrence and non-occurrence. It was assumed that the item occurring the least times per hour of observations conducted in the RCT (*n *= 175, 647.5 h) would also be the least occurring item per hour in the reliability evaluation study. We chose the least occurring item within the category of ‘ergonomic exposures’ as this category was our main interest. In the RCT observations, this item was ‘kneeling’, which occurred 1.35 times per hour of observation. The calculation showed that within 30 h of observation, ‘kneeling’ would occur at least 27 times with a probability score of 98%. This was considered sufficient to perform inter-rater reliability evaluations. The likelihood that our raters would be exposed to a better approximation of the mean exposure and variance of the childcare worker population would increase with an increased number of observed childcare workers [33]. Thus, to diminish individual worker bias we divided the 30 h of observation needed, into as many participants as were practically and logistically feasible. We contacted three day nurseries who were all willing to participate. From each nursery, five childcare workers volunteered and were observed for two hours each. 

#### 2.2.2. Raters and Rater Training

Prior to performing the observations, raters had approximately 8.5 h of training following a five-step process presented in Table 2. 

#### 2.2.3. Observation Procedures

Raters were instructed to explain the purpose of observation to the childcare worker and to request him/her not to alter work tasks while being observed. The raters were instructed not to interact with the childcare worker and children while observing, and also to stay in the background as much as possible without limiting their chance of observing the childcare worker properly. If the childcare worker asked for privacy, e.g., to tuck in a child that easily got distracted by the rater or to have a private conversation with a parent, the raters were instructed to register the item Rater/observation break (item no. 19) and wait until the childcare worker came back. The raters were also instructed not to talk to each other while conducting observations. For items number 3–5 (Table 1), raters were instructed to wait 2–3 s in order to decipher if the activity should be registered as Carry (item no. 3), Lift (item no. 4) or Push, pull or partial lift (item no. 5).

#### 2.2.4. Inter-Rater Reliability

The reliability evaluation was conducted as an inter-rater reliability (IRR) evaluation between two raters. The IRR was evaluated as the agreement of either instantaneous occurrence (i.e., occurring at the same time point), onset of duration (i.e., same starting point) or simultaneous duration (i.e., agreement between raters that the item occurred in the same time period—not the total duration) of items. As argued in Section 2.2.1, an item needed to be registered at least 27 times in total to be considered sufficient for reliability evaluation. A total number of registrations or duration in seconds below this number therefore disqualified IRR evaluation. Agreement was determined for all items including descriptive factors, but as the purpose of our study was to assess the overall reliability of the TRACK instrument, agreement for descriptive factors are not presented. To display agreements and disagreements and to inspect considerable misclassifications of items, a confusion matrix of all registered pair of observations was generated (not shown) [34]. The matrix revealed severe misclassification between items ‘Lift’ (item no. 4) and ‘Push, pull, partial lift’ (item no. 5), thus the two items were collapsed into a new item termed ‘Manual handling’. 

#### 2.2.5. Data Collection

Data was collected over the course of four weeks in 2017 in three day nurseries in Copenhagen. For data collection, each rater had a tablet with the Pocket Observer software installed. An observation began by both raters jointly counting down from three to ensure the exact same start time. Both raters then closely followed the childcare worker and registered all events occurring according to the manual. After exactly 120 min, the two raters jointly counted from three again to get the same stopping time. Tablets were then connected to a computer for download and storage of data on a secure server. Throughout the data collection period, tablets were regularly synchronized to the same main computer, which secured the same absolute time of the tablets. 

### 2.3. Data Analyses

#### 2.3.1. Data Processing

Data from the tablets was imported into ‘The Observer XT’ software (The Observer XT version 14; Noldus Information Technology, The Netherlands) and IRR expressed as strength of agreement was determined using the in-built inter-rater agreement determination function, and manually checked for adherence to protocol. As the start time of the observations were per protocol synchronized by the rater pair (see Section 2.2.5), absolute time was used for IRR examinations. Agreement of both item and start time between raters was then evaluated using the software. The outputs for each item were ‘Rater agreement’, ‘Rater A registers’ or ‘Rater B registers’ calculated as [A_yes_, B_yes_], [A_yes_, B_no_] and [A_no_, B_yes_], respectively. As we did not have video recordings of the observations to act as a ‘golden standard’ measurement because of ethical reasons, we could not assume that both raters were correct when they did not register anything, therefore [A_no_, B_no_] always equals zero. Therefore, the Cohen’s Kappa coefficient is not suitable for the inter-rater agreement calculations in this study. Instead, Gwet’s Agreement Coefficient (AC_1_) [35,36] was used to determine the inter-rater agreement as this does not include the no-registration [A_no_, B_no_] data. A 2x2 contingency table was constructed for each investigated item. 

For each event, the software’s inter-rater agreement determination function checked each synchronized timestamp via rolling-window analysis of the time-series data for agreement between the two raters on each individual event. Agreement was determined in four different ways depending on the type of event (state or point) and type of entity (‘instantaneous occurrence or ‘onset of duration’ and ‘duration’) as exemplified in Figure 1. Agreement of events with the entity types ‘duration’ (i.e., mutually exclusive and exhaustive state event items or non-exhaustive type of state event) were determined on a second-by-second basis (Figure 11,2). Agreement of events with the entity type ‘instantaneous occurrences’ (i.e., point event items) or ‘onset of duration’ (i.e., non-exhaustive state event items) was determined if both raters had registered the onset of event within a 5-second tolerance window (Figure 13,4). The tolerance window algorithm calculated all possible combinations of event agreements within the specified tolerance window and presented the optimal agreement configuration.

#### 2.3.2. Statistical Analyses

Agreement between raters was evaluated using percentage agreement as well as Gwet’s AC_1 _agreement coefficient [35,37]. Equations for calculation of the latter are presented in Appendix A. The agreement coefficient was interpreted by the Landis and Koch benchmark scale [38], as recommended by Gwet, e.g., as the intervals are more narrow than in the Fleiss’ benchmark scale [37]. The scale has six intervals: <0.0 (poor), 0.00 to 0.20 (slight), 0.21 to 0.40 (fair), 0.41 to 0.60 (moderate), 0.61 to 0.80 (substantial) and 0.81 to 1.00 (almost perfect) [38]. Percentage agreement between raters was calculated from number of scorings where rater A agrees with rater B (^b ^in Table 3) divided by the total number of scorings where at least one rater had recorded the particular item (^d ^in Table 3). Output from the in-built inter-rater agreement determination function in the Observer XT software (The Observer XT version 14; Noldus Information Technology, The Netherlands) was exported to Microsoft Excel (Microsoft Office Professional Plus 2016, Washington, USA 2010) to calculate percentage agreement and agreement coefficients.

### 2.4. Ethical Considerations

Recording video and using it as a gold standard (truth) to which rater observations could be compared was not an option as the privacy of the children needed to be protected. The National Research Centre for the Working Environment has an institutional agreement with the Danish Data Protection Agency about procedures to treat confidential data (journal number 2015-41-4232), such as by securing data on a protected drive with limited access and making all individual data pseudonymous. The regional ethics committee of Frederiksberg and Copenhagen under the Danish National Committee on Biomedical Research Ethics evaluated a description of the TOY-project and concluded that, according to Danish law as defined in Committee Act § 2 and § 1, the study is exempt from approval from the local ethics committee (reference number 16048606).

## 3. Results

The IRR of the TRACK instrument was generally good. As presented in Table 3**,** a total of twenty IRR calculations (percentage agreement and Gwet’s Agreement Coefficient (AC_1_)) were conducted on sixteen unique items with four items being evaluated with two entity types. These four items were Squat, Kneel, Carry and Sit on floor. IRR evaluation showed agreement percentages ranging from 42.4 to 99.5 with corresponding AC_1_s ranging from 0.02 to 0.99. Nineteen AC_1_s were at least 0.21 which according to the benchmark scale proposed by Landis and Koch is ‘fair’ [38]. Nine were classified as almost perfect’, four as ‘substantial’, four as ‘moderate’, two as ‘fair’ and one as ‘slight’. In total, 17 (85%) of evaluated items had moderate to almost perfect AC_1_s and agreement percentages higher than 60%. 

### 3.1. Inter-Rater Reliability of Settings

For settings, IRR evaluation showed ‘almost perfect’ agreement for Indoor (AC_1 _of 0.82) and ‘substantial’ agreement for Outdoor (AC_1_ of 0.79).

### 3.2. Inter-Rater Reliability of Ergonomic Exposures

The items Squat, Kneel, Carry, Sit on floor, Manual handling and Acute strain were evaluated with different entity types as according to their type of event (^b^ in Table 3). IRR evaluation was possible for nine items of which seven had AC_1_s ranging from 0.70 to 0.97 classifying them as ‘substantial’ or ‘almost perfect’. The remaining two (Squat—duration and Manual handling—instantaneous occurrence) had ‘fair’ AC_1_s at 0.33 and 0.23, respectively. When comparing items evaluated with two different entity types, AC_1_s were similar for Kneel (0.70 and 0.85), Carry (0.82 and 0.71) and Sit on floor (0.85 and 0.97). For Squat, AC_1_s were 0.71 and 0.33 for onset of duration and duration, respectively.

### 3.3. Inter-Rater Reliability of Work Situations

Nine items within the work situations category could be IRR evaluated. AC_1_s for work situations were generally high except Hand-over and pick-up which had an AC_1_ of 0.02 (‘slight’). Of the remaining eight work situation items evaluated, four had AC_1_s classifying them as ‘almost perfect’: Diaper and/or clothes change (0.84), Eat and/or group gathering (0.86), Outing (0.99) and Other childcare work (0.83). The last four situation items had AC_1_s classifying/interpreting them as ‘moderate’: Other clothes change (0.41), Tuck children up or pick up when awake (0.53), Clean up, tidy up or preparation (0.42) and Rater/observation break (0.61)

### 3.4. Inter-Rater Reliability of Disturbances

A disturbance was only registered three times in total, precluding IRR evaluation.

## 4. Discussion

We investigated the reliability of a newly developed observational instrument for assessing ergonomic work demands during childcare work—TRACK—and the IRR was generally good. Except for the three items with fair or slight agreement, evaluated items showed at least moderate agreement, and 13 out of 20 items showed almost perfect or substantial agreement. 

### 4.1. Inter-Rater Reliability between Different Items

Agreement coefficients for evaluated items varied between 0.02 and 0.99, with greatest variation for items in the category of work situations. The items in the categories of ergonomic exposures and settings generally had high agreement. This variation in inter-rater agreement between the different categories may be explained by their different features: items within the ergonomic exposure category describe physical movements while items within the work situation category describe conditions or localisations. Settings are limited to two exhaustive options. We wanted to determine continuous agreement between the raters (i.e., that they at any time point scored the same items). This puts a high emphasis on the synchronicity of the raters and the subsequent temporal timeline of the observational data. An initial sensitivity test of the tolerance window (data not shown) showed an increased number of agreements until a tolerance window size of 10 s, but with the instruction to wait 2–3 s to register an item with an addition of 1–2 s of reaction time, meant we could not justify a tolerance window larger than 5 s. Additionally, determining the agreement based on the total duration or total number of observed events would likely have produced stronger agreements overall. 

The existing literature on assessment of ergonomic work exposures for childcare has not been tested for reliability [23,24,25,26,27,28,29]. Comparison of our findings is therefore limited to similar observation instruments for different occupational groups. Karstad et al. developed and evaluated, without use of video recordings, a similar observation instrument for use in eldercare work and found generally good inter-rater reliability [39]. Only one item of the instrument resembles an item of the TRACK instrument, i.e., squat. AC_1_ for squatting was 0.75, which is very similar to our AC_1_ at 0.71 (Table 3). Comparing our findings with other observation methods evaluated with video recordings is restricted as our study design was in-field. However, our design resembles that of Village et al. who evaluated inter-rater reliability of an observation instrument for documenting physical exposures to back injury risk factors (Back-EST) and found generally good to very good agreement between raters although direct comparison is precluded as the agreement was expressed as intraclass correlations [18].

#### 4.1.1. Ergonomic Exposures

The strength of agreement was ‘substantial’ to ‘almost perfect’ for the ergonomic exposures except for Manual handling and Squat-duration, which was ‘fair’. Agreement for items in this category can be explained by the distinct motions of relatively large body segments, and therefore being more easily identified and scored correctly [6]. Manual handling comprises ergonomic exposures with smaller and more spontaneous/rapid movements (lift, push, pull and partial lift), which are more challenging to identify and record [6], thus likely to cause weaker agreement between the raters. Furthermore, the observation manual lacked a description of continuously push/pull (e.g., of pram or swing), which may have resulted in different interpretations and scorings between the raters. For future use of TRACK we suggest to provide a better description and distinction of the items Lift (no. 4) and Push, pull, partial lift (no. 5) to the observation manual and rater training to minimise confusion between these two items and subsequent merging into Manual handling. When evaluating agreement of duration of ergonomic exposures, Squat had a noticeably lower AC_1_ than Kneel, Carry and Sit on floor. This might be explained by the large number of rather short observations for Squat with a mean duration of 17 s vs. mean durations above 50 s for Kneel, Carry and Sit on floor. This puts increased emphasis on the exact timing of the raters for starting and stopping the observation synchronically for achieving high agreement. The variation in agreement between the two entity types for ergonomic exposures (onset of duration/instantaneous occurrence vs. duration) generally indicates:It is easier to correctly observe the occurrence of easily recognisable and well-defined items involving large body parts or movements such as Squat, Carry and Sit on floor vs. more poorly defined short-lasting finicky items such as Manual handling.It is easier to correctly observe the duration for longer lasting items such as Sit on floor vs. shorter lasting items such as Squat.

However, evaluating items with both entity types gives highly valuable information as both frequency and duration of ergonomic exposures greatly impacts the total ergonomic work demand. For increased reliability for future use of TRACK we suggest to add better examples to the observation manual of what squatting looks like in real-life childcare work. However, it is noticeable that agreement for Squat—onset of duration was ‘substantial’ while Squat—duration was only ‘slight’. This shows that the raters were able to detect the movement but did not agree for how long the movement was sustained or started/ended. This can best be explained by the short tolerance window of five seconds rather than difficulties in detecting squat, and the relatively short mean duration of Squat as mentioned above. 

#### 4.1.2. Work Situations

The strength of agreement was ‘moderate’ to ‘almost perfect’ for all work situations except Hand-over and pick-up, which was ‘slight’. Some work situations occur in a specific room (e.g., Diaper and/or clothes change in the bathroom and Other clothes change in the wardrobe) or with the use of specific objects (e.g., broom or bucket of water for Clean-up, tidy up or preparation situation) and are thereby items feasible to identify, ensuing high agreement. The weak agreement of Hand-over and pick-up might be due to the situation being difficult to predict, occurring in all settings, often occurring simultaneously with other situations, and being split into short periods. With this in mind, we are not surprised that this particular situation has lower agreement than the others, especially as determination of agreement in this study design did not only require agreement between raters on total duration, but also synchronisation of start- and stop-times. However, to increase reliability for ‘Hand-over and pick-up’ we suggest to separate it into ‘Hand-over’ and ‘Pick-up’, and to add better explanations of these situations in the manual.

#### 4.1.3. Settings

The strengths of agreement were ‘almost perfect’ and ‘substantial’ for Indoor and Outdoor, respectively. In theory, the setting should be easy to decipher and score correctly as the number of options are limited to two, exhaustive items. However, in reality, only Indoor showed an AC_1 _of 1.00. In practice, childcare workers were mainly outdoors on two occasions: Care for children in the playground or when they pushed prams with sleeping babies in or out from the nursery (as it is a natural thing in Denmark to place a baby outside in a pram during naps). The latter had short durations, and scoring of the Outdoor item was therefore influenced by the raters’ reaction time which had an effect on the strength of agreement. This hypothesis is substantiated by the very similar durations of when only either rater A or B registered Outdoor (263 s for rater A, 249 s for B). In other words, they agreed on the total duration of Outdoor, but were not completely aligned on the time of occurrence. 

To the best of our knowledge, this is the first observational instrument developed to assess ergonomic work demands in childcare work with evaluation of the inter-rater reliability, and the study’s results are thus not directly comparable with other studies. However, the inter-rater reliability of this observational instrument is similar to the inter-rater reliability of a very comparable observational instrument developed for eldercare workers [39]. 

### 4.2. Applicability of the TRACK Instrument for Assessing Ergonomic Work Demands in Childcare Work

Knowledge about ergonomic work demands during childcare is very limited. The knowledge available is predominantly based on self-report or resource-intensive direct objective measurements with simultaneous video-recording [23,24,25,26,27,28,29]. Thus, the TRACK instrument is needed for both researchers and practitioners. Popular objective methods, such as accelerometers, are able to measure some ergonomic work exposures (e.g., walking and running), postures (e.g., sitting and standing) and movements (e.g., forward bending and arm elevation) over longer time-periods at work and leisure. However, they are currently not well suited for other types of ergonomic work demands like carrying, acute strain and social/contextual situations [40,41]. Therefore, we see a great need and usability of the developed TRACK instrument for researchers to provide a better knowledge fundament of the ergonomic work demands during childcare work, and for qualifying preventive interventions for reducing ergonomic exposures among childcare workers by occupational practitioners. As 17 of 20 evaluated items showed ‘moderate’ to ‘almost perfect’ agreement, we consider the TRACK instrument to be reliable for both researchers and practitioners to use for assessing the ergonomic work demands during childcare work. The different entity types that items are evaluated with allow detailed assessment compared to only assessing totals for each item. This provides a knowledge base that can be analysed for individual childcare workers or summed up to group level to assess amplitude of ergonomic work demands. In addition, the instrument contextualises the ergonomic exposures of childcare work to the situations and settings in which they occurred. In addition, no particular educational background (e.g., occupational therapist) of the observer is necessary and no advanced instruments other than a handheld device like a tablet and a computer are required. The TRACK instrument is designed and evaluated for the use in day nurseries, but could potentially also be used with minor modifications in jobs with similar settings and work tasks (e.g., patient care). 

### 4.3. Strengths and Limitations

In the literature, rater training has been described as being of significant importance for achievement of high inter-rater reliability in observations of ergonomic exposures [6]. The raters in this study received thorough training and had almost one hundred hours of observation experience. Additionally, different employees in different nurseries were observed during real-time normal settings which strengthen the external validity of the study. 

As the observations occurred during normal settings, some items (Acute strain, Childcare worker break, Disturbance) were too rarely observed to evaluate the inter-rater reliability. However, we did not expect either Acute strain or Disturbance to occur sufficiently for IRR evaluations as they also had very low frequencies in the RCT study. As all other items, especially in the category of ergonomic exposures, occurred sufficiently for IRR evaluation, we feel confident that our calculation of hours of observation required for evaluating inter-rater reliability was true. Interestingly, Kneeling was not the least occurring ergonomic exposure item in the reliability evaluation study. The frequency of kneeling in the RCT study was 1.35 times per hour while the average rater frequency in this study was 1.83 times per hour. Both Carry and Sit on floor had lower frequencies in the reliability study (average between raters at 1.70 and 1.76, respectively). However, both of these items had sufficient numbers of occurrence for IRR evaluation. Conducting observations of ergonomic work demands in childcare in real-life settings will always be a reflection of the individuality among participants and e.g., their physical capacity, workplace ergonomic culture, physical environment they are working in, education and experience etc. 

A limitation is the study’s inter-rater reliability being calculated based on scorings from only one observer pair and might therefore have limited generalizability to other observers. Video recordings could have provided true information about agreements of an event. Video recordings could also have allowed multiple pairs of raters. However, video recordings of children in nurseries are ethically challenging, and it was not possible to retrieve permission for video recordings in this study. Training sessions that included video recordings could also have increased strengths of agreement, especially for the fuzzy situations like hand-over and pick-up that are not defined by a certain location or time and not always have a short and precise occurrence. Parents sometimes chose to hang out in the nursery for a little while which affected precise registrations of this item. Video recordings of multiple hand-over and pick up situations and plenum discussions of correct start and stop time could have trained raters to better decipher and register this item. Collapsing the items Lift and Push, pull and partial lift into Manual handling is also a weakness of our study. The initial confusion matrix showed substantial misclassification of the two items, which requires merging of the single items. However, we do acknowledge that essential information is lost due to lifting, pushing and pulling are important ergonomic work exposures in childcare [2,3,4]. In a study with a very similar observation instrument, it was possible to register the items Lifting and Pushing/pulling separately [39]. However, this study was conducted in elderly care work where the work tasks and behaviours are likely to occur in a more planned and distinct manner than in the care for small children aged 0 to 3 years. We suggest to improve both definitions and guidance for registration of the these items before using TRACK, as recommended by Kilbom [15] as a way to improve reliability of observations. A last limitation is the tolerance window of 5 s we allowed for items evaluated for onset or instantaneous occurrence. This was however chosen as we assumed a reaction time of 1–2 s to intercept and register an item and because instructions for items no. 3–5 asked raters to wait 2–3 s to decide which of the three items should be registered. 

## 5. Conclusions

The inter-rater reliability of the TRACK instrument for childcare work was generally high. Out of 20 evaluated items, 17 items (85%) could be observed with ‘almost perfect’ to ‘moderate’ inter-rater reliability. These findings support that the TRACK instrument is well suited for assessing ergonomic work demands in childcare, particularly as a supplement to device-based measurements, such as accelerometers. Observations provide contextualization of the ergonomic exposures of childcare to situations and settings in which they occurred, providing an extra layer of useful information. Overall, the TRACK instrument is considered to be reliable for both researchers and practitioners to use for assessing occurrence and duration of ergonomic work demands during childcare work. 

## Figures and Tables

**Figure 1 ijerph-17-01607-f001:**
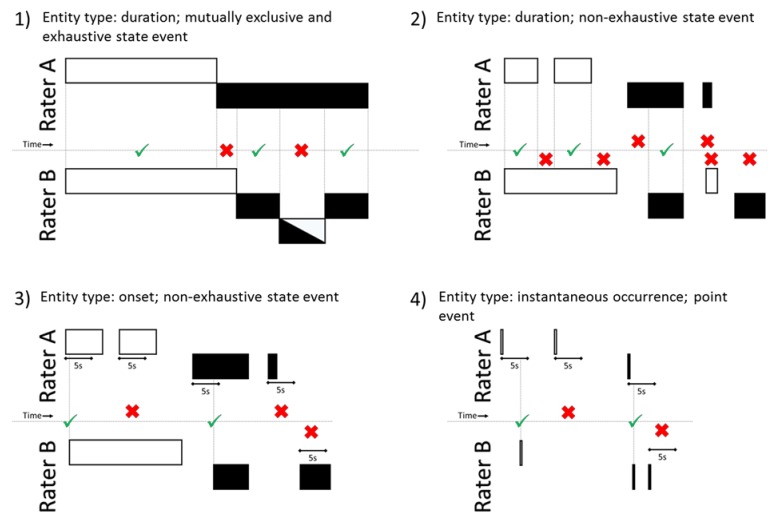
Examples of agreement determination between rater A and rater B for different entity types (instantaneous occurrence, onset of duration or duration). A check sign represents agreement, while a cross mark represents periods of non-agreement between the two raters. The color-coded boxes in part 1, 2 and 3 represent registered state events, while the lines in part 4 represent single point events. Recorded time goes from left to right. (1) Agreement on duration of mutually exclusive and exhaustive type of state event: Agreement was determined second-by-second for each event. (2) Agreement on duration of non-exhaustive type of state event: Agreement was determined second-by-second for each event. (3) Agreement on onset of duration of non-exhaustive type of state event: Agreement was determined if the two raters registered onset of the event within a 5-second tolerance window and calculated as [A_yes_, B_yes_]. If only one rater had registered onset within the 5-second tolerance window, it was determined as the corresponding [A_yes_, B_no_] or [A_no_, B_yes_].(4) Agreement on instantaneous occurrence of point events: Agreement was determined if the two raters registered the point event within a 5-second tolerance window and calculated as [A_yes_, B_yes_]. If only one rater had registered the point event within the 5-second tolerance window, it was determined as the corresponding [A_yes_, B_no_] or [A_no_, B_yes_].

**Table 1 ijerph-17-01607-t001:** Overview of the observaTion woRk demAnds Childcare worK (TRACK) observation instrument for assessing ergonomic work demands in childcare work. The instrument consists of 20 items in four categories. Six items contain descriptive factors as sub-items.

Item	Definition	Item Number	Descriptive Factor ^a^	Definition	Type of Event ^b^
Settings					
Indoor	Being indoors	1			State event—mutually exclusive and exhaustive
Outdoor	Being outdoors	2			State event—mutually exclusive and exhaustive
Ergonomic exposures					
Carry	Child or object not in contact with the surface and carried for at least 2–3 s and/or two steps	3	Large/heavy weight child	Rule of thumb: Child able to walk by itself and/or weighs above 12 km	State event—non-exhaustive
			Small/light weight child	Rule of thumb: Child not able to walk by itself and/or weighs below 12 km	
			Other	Objects. Triviality level: 2–3 km	
Lift	Child or object not in contact with the surface but carried for less than 2–3 s and/or two steps	4	Large/heavy weight child	Rule of thumb: Child able to walk by itself and/or weighs above 12 km	Point event
			Small/light weight child	Rule of thumb: Child not able to walk by itself and/or weighs below 12 km	
			Other	Objects. Triviality level: 2–3 km	
Push, pull, partial lift	Child or object moved or supported without he/she/it losing contact to the existing surface	5	Large/heavy weight child	Rule of thumb: Child able to walk by itself and/or weighs above 12 km	Point event
			Small/light weight child	Rule of thumb: Child not able to walk by itself and/or weighs below 12 km	
			Other	Objects. Triviality level: 2–3 km	
Sit on floor	Sitting either on the bare floor or on a thin mattress (<5 cm) with the bodyweight on at least one buttock, and feet and buttock(s) in approximately the same height. This also includes sitting cross-legged or in a mermaid position.	6	With sit pad (only)	CW sits on cushion or pad that is approximately 5–15 cm thick when not being loaded	State event—non-exhaustive
			With back support (only)	CW leans against something	
			With sit pad + back support	CW sits on cushion or pad that is approximately 5–15 cm thick when not being loaded + leans against something	
			Without any support	No cushion or pad or something to lean against	
Squat	Sitting position where neither knees nor buttocks touch the floor or ground surface and the angle of the knees is ≤90 degrees	7			State event—non-exhaustive
Kneel	Knee(s) and lower leg(s) are in contact with the floor or ground surface. Heel(s) can, but do not have to, touch the buttock(s)	8	With pad	Cushion or pad that is approximately 5–15 cm thick when not being loaded	State event—non-exhaustive
			Without pad	No cushion or pad	
Acute strain	An unforeseen incident with a sudden physical strain on the CW, e.g., if a tripping child is caught to cushion or prevent a fall	9			Point event
Situations					
Hand-over and pick-up	CW interacts with parent(s) (or similar responsible adult/adolescent) upon arrival or leaving. Hand-over or pick-up of a child can be recorded as several situations if interrupted or split	10			State event—mutually exclusive and exhaustive
Diaper and/or clothes change	All diaper and/or clothes changes done in the ward or the child changing facility adjoining each ward	11			State event—mutually exclusive and exhaustive
Other clothes change	All other situations with change of clothes not done in the ward or baby changing facility, e.g., putting on outdoor clothes in the wardrobe	12			State event—mutually exclusive and exhaustive
Eat and/or group gathering	Organised gatherings of all (awake) children, e.g., when eating lunch or doing a group activity	13			State event—mutually exclusive and exhaustive
Tuck children up or pick up when awake	Tucking children up and picking them up when they wake up, either in cradle or dormitory	14			State event—mutually exclusive and exhaustive
Clean-up, tidy up or preparation	Work related to cleaning, tidying, and preparing activities or similar, with or without accompanying children	15			State event—mutually exclusive and exhaustive
Outing	Field trips outside the nursery’s cadastral plot	16			State event—mutually exclusive and exhaustive
Other childcare work	Remaining work not comprised in situations with item number 10–16, 18 or 19	17			State event—mutually exclusive and exhaustive
Childcare worker break	Breaks for the CW planned in his/her work schedule, e.g., scheduled lunch breaks or meetings	18			State event—mutually exclusive and exhaustive
Rater/observation break	Periods where the rater was unable to observe the CW, e.g., for privacy reasons, unplanned breaks or due to impediments of the furnishing	19			State event—mutually exclusive and exhaustive
Disturbances					
Disturbance	Pronounced and acute interruption of the CW’s current task	20	Colleague	Other CWs	State event—non-exhaustive
			Parent	Parent or similar responsible adult/adolescent	
			Child	Other children in nursery	
			Other	Objects, e.g., a phone ringing	

CW: childcare worker; ^a ^descriptive factors are modifiers to an item, i.e., sub-items. If an item has descriptive factors one factor must always be registered. ^b ^Type of event defines the sampling strategy. ‘Point event’ is registered at a single time point providing information of the instantaneous occurrence of an event. ‘State event—start–stop’ is registered over time using manual start- and stop-time providing information of both the instantaneous occurrence and duration of an event. ‘State event—mutually exclusive and exhaustive’ is registered over time using manual stop-time and automatic stop-time by registration of another item within the same category. This type of event also provides information of both the instantaneous occurrence and duration of an event.

**Table 2 ijerph-17-01607-t002:** Overview of the training of the raters. The rater training was completed over two weeks before the raters were to conduct observations out at workplaces.

Session	Content	Duration
1	All raters received the observation manual and the observation protocol and were instructed to read both carefully.	1 h
2	Lab session: The researchers who had participated in developing the TRACK instrument, instructed the raters in the practical and technical use of the instrument.	2 h
3	Field session: In pairs of two, the raters observed one childcare worker simultaneously for 45 consecutive minutes. The raters were not allowed to talk or show their tablets to each other. Subsequently, the observation was evaluated, comparing ratings and discussing challenges and perceptions of item definitions by completing an assessment form. The two steps were repeated three times in total.	3 h
4	Lab session: All raters and researchers verbally evaluated the use of the TRACK instrument based on the field experiences and revised the observation manual to clarify and correct differences in how the item definitions were interpreted.	1.5 h
5	All raters received the revised observation manual and were instructed to read it carefully.	1 h

Two raters performed the observations for the reliability evaluation. Prior to the reliability evaluation, both raters had roughly one hundred hours of field experience with TRACK. Rater A had performed 26 observations and rater B 25 observations, with each observation lasting approximately four hours [1]. Rater A was male and 25 years old; rater B was female and 27 years old.

**Table 3 ijerph-17-01607-t003:** Inter-rater reliability of ergonomic work demands items in the TRACK observational instrument, expressed as the percentage of agreement (%) between rater A and B, agreement coefficient (Gwet’s AC_1_) and strength of agreement. The item Manual handling is generated from the items Lift (item no. 4) and Push, pull, partial lift (item no. 5).

Category and Item	Entity Type	Item Number ^a^	Rater A Agrees with Rater B ^b^	Only Rater A Scores^ c^	Only Rater B Scores ^c^	Total Scorings ^d^	% Agreement ^e^	Gwet’s AC_1 _^f^	Strength of Agreement ^g^
Settings									
Indoor	Duration (seconds)	1	105,182	253	272	105,706	99.5	1.00	Almost perfect
Outdoor	Duration (seconds)	2	2374	263	249	2887	82.2	0.79	Substantial
Ergonomic exposures									
Squat	Onset of duration (no. of events)	7	118	16	20	154	76.6	0.71	Substantial
Kneel	Onset of duration (no. of events)	8	84	13	13	110	76.4	0.70	Substantial
Carry	Onset of duration (no. of events)	3	86	5	11	102	84.3	0.82	Almost perfect
Sit on floor	Onset of duration (no. of events)	6	92	6	8	106	86.8	0.85	Almost perfect
Squat	Duration (seconds)	7	1507	453	723	2683	56.2	0.33	Fair
Kneel	Duration (seconds)	8	5029	410	341	5780	87.0	0.85	Almost perfect
Carry	Duration (seconds)	3	4246	583	689	5537	76.7	0.71	Substantial
Sit on floor	Duration (seconds)	6	20,222	273	421	20,915	96.7	0.97	Almost perfect
Manual handling	Instantaneous occurrence (no. of events)	4 + 5	196	116	71	383	51.2	0.23	Fair
Acute strain ^h^	Instantaneous occurrence (no. of events)	9	0	1	1	2	N/A	N/A	N/A
Work situations									
Hand-over and pick-up	Duration (seconds)	10	1258	875	836	2969	42.4	0.02	Slight
Diaper and/or clothes change	Duration (seconds)	11	9669	537	1048	11,254	85.9	0.84	Almost perfect
Other clothes change	Duration (seconds)	12	1485	919	72	2476	60.0	0.41	Moderate
Eat and/or group gathering	Duration (seconds)	13	20,879	2197	820	23,896	87.4	0.86	Almost perfect
Tuck children up or pick up when awake	Duration (seconds)	14	5493	2397	437	8327	66.0	0.53	Moderate
Clean-up, tidy up or preparation	Duration (seconds)	15	6137	2642	1362	10,141	60.5	0.42	Moderate
Outing	Duration (seconds)	16	191	1	1	193	99.0	0.99	Almost perfect
Other childcare work	Duration (seconds)	17	50,621	1684	6847	59,152	85.6	0.83	Almost perfect
Childcare worker break ^h^	Duration (seconds)	18	0	0	0	0	N/A	N/A	N/A
Rater/observation break	Duration (seconds)	19	831	254	90	1176	70.7	0.61	Moderate
Disturbances									
Disturbance ^h^	Onset of duration (no. of events)	20	1	2	0	3	N/A	N/A	N/A

^a^ See Table 1. ^b ^The total number of registrations of an item during all observations when raters agreed. ^c ^The total number of registrations of an item during all observations by each rater when raters disagreed. ^d ^The total number of registrations of an item during all observations (b + c). ^e^ % of agreement between raters (b/d × 100). ^f ^Gwet’s AC_1_ coefficient of inter-rater reliability [35,37]. ^g^ Classification as according to Landis and Koch [38]. ^h^ Items with insufficient number of total scorings to allow inter-rater reliability examinations.

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
