# Peer review of "Inter-Rater Reliability of Ergonomic Work Demands for Childcare Workers Using the Observation Instrument TRACK"

_ijerph, 2020, doi:10.3390/ijerph17051607_

Round 1

Reviewer 1 Report

MAIN POINTS

The paper is very clear in its objectives and the methodology section allows to affirm that the results are based on a scientific methodology. In general, I would suggest to the authors just to strengthen the literature bases of the introduction and discussion sections. The paper is based on a low number of references (32). Introduction section could be more based on literature. In the discussion section, it seems that the discussion is between the authors and their results. I didn´t find a good discussion with other authors. I hope in the next version to have another 20-30 references from other authors. In addition, there are repetitions of content between the introduction and the discussion sections (Example: The text in Line 346 ("The knowledge available is predominantly based on self-report, which can be both imprecise and biased [5-7]") is similar to the one in line 36 ("The existing knowledge of ergonomic work exposures of childcare workers is limited to self-reported measures, which are known to be imprecise and potentially biased [5-8]"). I'm expecting different texts between introduction/literature and discussion).

In the discussion section, why do you consider the moderate result (0.41 to 0.60) as positive in the evaluation of "Applicability of the TRACK instrument"?

Line 363: "In addition, no particular education or background of the observer is necessary and no advanced instruments other than a handheld device like a tablet and a computer are required". I understand that raters don't need a previous specific  knowledge, but at least a lot of training.

In the discussion I expected to receive suggestions for TRACK improvements in order to improve the results of the items that had an evaluation between "poor" and "moderate"

MINOR POINTS

Line 69, 120, 144, ...: "Error! Reference source not found.."

Line 76, 153, ...: section 0."

Line 76 I think section 0 is section 2.3.1. But I would like to see this image next to this text.

Line 92-93: "To evaluate reliability, 30 hours of concurrent field observations of childcare work was 92 conducted by two raters." I suggest to include "simultaneously" at the end of the sentence.  

Section "2.2.2. Raters and rater training": I suggest leaving the information about training raters in the table OR in the text. The information is repeated.

Line  219: "Supplementary File 1". I didn´t find it.

Line 407: "Kilbom (1994) [32]". I suggest to write " Kilbom [32]"

Author Response

Comments and Suggestions for Authors

MAIN POINTS

The paper is very clear in its objectives and the methodology section allows to affirm that the results are based on a scientific methodology. In general, I would suggest to the authors just to strengthen the literature bases of the introduction and discussion sections. The paper is based on a low number of references (32). Introduction section could be more based on literature. In the discussion section, it seems that the discussion is between the authors and their results. I didn´t find a good discussion with other authors. I hope in the next version to have another 20-30 references from other authors.

Author comment:
Thank you for your comment. We agree that the literature in the introduction section can be expanded, and have included more references to support the argumentation for the need of the study. All changes can be seen with Track Changes and new references have been highlighted in comments.
Comparing our results to other studies is very restricted as no observational instruments specifically for childcare have been evaluated for (inter-rater) reliability. The next step would therefore be to compare with studies with similar observational instruments for other occupational groups and/or similar study designs. This, however, is very restricted as almost all inter-rater reliability evaluations are designed as rating against video recordings which is a completely different setup than in-field observations with two raters who naturally cannot stand on top of each other and monitor the same task from the identical point of view. Our raters had to “fit in” the daily work of those observed. Futhermore, comparisons only make sense if the items evaluated are more or less identical to those in our instrument, TRACK. We have, however, included some more relevant references for the discussion of our results.

In addition, there are repetitions of content between the introduction and the discussion sections (Example: The text in Line 346 ("The knowledge available is predominantly based on self-report, which can be both imprecise and biased [5-7]") is similar to the one in line 36 ("The existing knowledge of ergonomic work exposures of childcare workers is limited to self-reported measures, which are known to be imprecise and potentially biased [5-8]"). I'm expecting different texts between introduction/literature and discussion).

Author comment:
Thank you for pointing this out. We sincerely apologise for this. The manuscript (introduction and discussion sections) has been revised to remove any repetitions.

In the discussion section, why do you consider the moderate result (0.41 to 0.60) as positive in the evaluation of "Applicability of the TRACK instrument"?

Author comment:
Thank you for bringing our attention to this. We do not consider ‘moderate’ on the Landis & Koch benchmark scale as a negative result. ‘Moderate’ (0.41 to 0.60) is the (at least or more than) sufficient interval, and it is the third highest rank, followed by ‘substantial’ (0.61 to 0.80) and ‘almost perfect’ (0.81 to 1.00). The bottom three intervals are ‘poor’ <0.0, ‘slight’ (0.00 to 0.20) and ‘fair’ (0.21 to 0.40). The ‘fair’ interval could also be interpreted as a positive (enough) finding but we believe the top three intervals to be high enough to conclude that the IRR was generally high as 85% of evaluated items had at least moderate AC1.

Line 363: "In addition, no particular education or background of the observer is necessary and no advanced instruments other than a handheld device like a tablet and a computer are required". I understand that raters don't need a previous specific  knowledge, but at least a lot of training.

Author comment:
We apologise for not being clear enough about this. We have revised the text to specify that we mean any specific educational background (e.g. occupational therapist). We agree that rater training is an essential part of being good at conducting observations using TRACK and to ensure high reliability of the observations.

In the discussion I expected to receive suggestions for TRACK improvements in order to improve the results of the items that had an evaluation between "poor" and "moderate"

Author comment:
Thank you for pointing this out. As we do not agree that items with moderate IRR is not a positive findings, we have added suggestions for improvements to the discussion section for the three items that scored below moderate: Squat – duration (line 353-359), Manual handling (line 336-339) and Hand-over and pick-up (line 370-372).

MINOR POINTS

Line 69, 120, 144, ...: "Error! Reference source not found.."
Author comment: We apologise for this and have corrected it although we suspect all of these cross reference errors occurred when the editorial office formatted the original document. They all worked in the uploaded uploaded manuscript.

Line 76, 153, ...: section 0."
Author comment: Thank you, it has been corrected.

Line 76 I think section 0 is section 2.3.1. But I would like to see this image next to this text.
Author comment: Thank you, it has been corrected. We feel Figure 1 sits better with section 2.3.1 to which it works as an visual explanation of the different ways data was processed to determine agreement. Mentioning Figure 1/section 2.3.1 in section 2.1.1 is only meant as a help to the reader for later referrals to “entity types”.

Line 92-93: "To evaluate reliability, 30 hours of concurrent field observations of childcare work was 92 conducted by two raters." I suggest to include "simultaneously" at the end of the sentence.  
Author comment: Thank you for pointing this out. We have followed your recommendation and removed the ‘concurrent’ and instead added ‘simultaneously’ in the end of the sentence.

Section "2.2.2. Raters and rater training": I suggest leaving the information about training raters in the table OR in the text. The information is repeated.
Author comment: Thank you highlighting this. The text now only refers to the table (Table 2) for explanation of rater training.

Line  219: "Supplementary File 1". I didn´t find it.
Author comment: We are sorry to hear you did not find the Supplementary File 1, presenting and explaining the equation for Gwet’s AC1. We can see that it was correctly uploaded together with the original manuscript.

Line 407: "Kilbom (1994) [32]". I suggest to write " Kilbom [32]"
Author comment: Thank you bringing our attention to this. The reference style has now been updated to that of the journal provided EndNote style (MDPI) and this should now be correct.

Reviewer 2 Report

Very professional and proficient; both in writing and in application of experimental method. I would be happy to work with this team and employ the methodology presented in my own work. Good understanding and presentation of the limitations of this method that will allow others to consider potential solutions to the problems identified.  

Things to address - 

Ln 69              : Error! Reference source not found..

Line 76           : Section 0

Ln 153            : Section 0

Ln 177            : Section 0

Ln 120            : Error! Reference source not found..

Ln 122            : while pinning uncertainties.” –

Does this mean noting / highlighting? Suggest: use a more commonly occurring synonym

Ln 125             : “in real-life setting” – Suggest: “in a real-life setting”

Ln 127             : See above (or make setting plural – settings).

Ln 144/145       : Error! Reference source not found..

Ln 153              : Section 0

Ln 177              : Section 0

Ln 224/225/226 : Error! Reference source not found..

Ln 231               :

“Video recordings was not an option to be used as a gold standard (truth) to which rater observations could be compared as the privacy of the children needed to be protected.”

Suggest: Recording video and using it as a gold standard (truth) to which rater observations could be compared was not an option as the privacy of the children needed to be protected.

Ln 237                : remove )

Ln 241/242         : Error! Reference source not found..

Ln 255/256         : Error! Reference source not found..

Page 13/20         : Error! Reference source not found..

Author Response

Comments and Suggestions for Authors

Very professional and proficient; both in writing and in application of experimental method. I would be happy to work with this team and employ the methodology presented in my own work. Good understanding and presentation of the limitations of this method that will allow others to consider potential solutions to the problems identified.  
Author comment: Thank you for your kind words. We would be happy to collaborate with you for any future data analysis or projects related to inter-rater reliability evaluation.

Things to address - 

Ln 69              : Error! Reference source not found..
Author comment: We apologise for this and have corrected it although we suspect all of these cross reference errors occurred when the editorial office formatted the original document. They all worked in the uploaded uploaded manuscript.

Line 76           : Section 0
Author comment: Thank you, it has been corrected.

Ln 153            : Section 0
Author comment: Thank you, it has been corrected.

Ln 177            : Section 0
Author comment: Thank you, it has been corrected.

Ln 120            : Error! Reference source not found..
Author comment: Thank you, it has been corrected.

Ln 122            : while pinning uncertainties.” –

Does this mean noting / highlighting? Suggest: use a more commonly occurring synonym
Author comment: Thank you for suggesting a better phrase. As reviewer 1 suggested to leave out either the text or table 2 of section 2.2.2, this sentence has now been deleted.

Ln 125             : “in real-life setting” – Suggest: “in a real-life setting”
Ln 127             : See above (or make setting plural – settings).
Author comment: Thank you for suggesting better phrasing of this. We chose to use the plural form and have corrected the text.

Ln 144/145       : Error! Reference source not found..
Author comment: Thank you, it has been corrected.

Ln 153              : Section 0
Author comment: Thank you, it has been corrected.

Ln 177              : Section 0
Author comment: Thank you, it has been corrected.

Ln 224/225/226 : Error! Reference source not found..
Author comment: Thank you, it has been corrected.

Ln 231               :

“Video recordings was not an option to be used as a gold standard (truth) to which rater observations could be compared as the privacy of the children needed to be protected.”

Suggest: Recording video and using it as a gold standard (truth) to which rater observations could be compared was not an option as the privacy of the children needed to be protected.
Author comment: Thank you. The sentence and has been rephrased and we agree it reads better now.

Ln 237                : remove )
Author comment: Thank you, it has been corrected.

Ln 241/242         : Error! Reference source not found..
Author comment: Thank you, it has been corrected.

Ln 255/256         : Error! Reference source not found..
Author comment: Thank you, it has been corrected.

Page 13/20         : Error! Reference source not found..
Author comment: Thank you, it has been corrected.
